# Guided Endodontics: Static vs. Dynamic Computer-Aided Techniques—A Literature Review

**DOI:** 10.3390/jpm12091516

**Published:** 2022-09-15

**Authors:** Diana Ribeiro, Eva Reis, Joana A. Marques, Rui I. Falacho, Paulo J. Palma

**Affiliations:** 1Dentistry Department, Faculty of Medicine, University of Coimbra, 3000-075 Coimbra, Portugal; 2Institute of Endodontics, Faculty of Medicine, University of Coimbra, 3000-075 Coimbra, Portugal; 3Center for Innovation and Research in Oral Sciences (CIROS), Faculty of Medicine, University of Coimbra, 3000-075 Coimbra, Portugal; 4Institute of Oral Implantology and Prosthodontics, Faculty of Medicine, University of Coimbra, 3000-075 Coimbra, Portugal

**Keywords:** access cavity, computer-aided, cone-beam computed tomography, dynamic navigation, endodontics, guided endodontics, printed template, root canal treatment

## Abstract

(1) Background: access cavity preparation is the first stage of non-surgical endodontic treatment. The inaccuracy of this step may lead to numerous intraoperative complications, which impair the root canal treatment’s prognosis and therefore the tooth’s survival. Guided endodontics, meaning computer-aided static (SN) and dynamic navigation (DN) techniques, has recently emerged as a new approach for root canal location in complex cases. This review aims to compare SN and DN guided endodontics’ techniques in non-surgical endodontic treatment. (2) Methods: an electronic search was performed on PubMed, Scopus, and Cochrane Library databases until October 2021. Studies were restricted by language (English, Spanish and Portuguese) and year of publication (from 2011 to 2021). (3) Results: a total of 449, 168 and 32 articles were identified in PubMed, Scopus, and Cochrane Library databases, respectively, after the initial search. Of the 649 articles, 134 duplicates were discarded. In this case, 67 articles were selected after title and abstract screening, of which 60 were assessed for eligibility through full-text analysis, with one article being excluded. Four cross-references were added. Thus, 63 studies were included. (4) Conclusions: guided endodontics procedures present minimally invasive and accurate techniques which allow for highly predictable root canal location, greater tooth structure preservation and lower risk of iatrogenic damage, mainly when performed by less experienced operators. Both SN and DN approaches exhibit different advantages and disadvantages that make them useful in distinct clinical scenarios.

## 1. Introduction

The first stage of non-surgical endodontic treatment is access cavity preparation. This crucial step must allow for a complete location of the root canal system, as well as its cleaning and shaping, in order to maximize microorganism elimination [1]. Inaccurately performing this stage may lead to numerous intraoperative complications such as missed root canals, root perforations, instrument fracture and/or weakening of the coronal structure, which impair the root canal treatment’s prognosis and therefore the tooth’s survival [2,3].

Teeth exhibiting pulp canal obliteration (PCO) or calcified canals often require a challenging [4] and time-consuming [5] endodontic treatment. PCO occurs by the deposition of hard tissue within the root canal space, thus making it narrower. It generally has no symptoms and can be detected via tooth discoloration and/or radiographical examination. This alteration can be due to chronic caries progression, previous vital pulp therapy procedures, tooth restoration or even luxation injuries [6,7]. Dental trauma, more commonly described in young patients, can instigate partially or entirely calcified root canals [8,9]. Moreover, cases of PCO due to the lifelong apposition of dentine are becoming more frequent as the number of elderly patients and their need for root canal treatment is increasing worldwide [1]. Endodontic therapy is indicated in 7% to 27% of PCO cases when apical periodontitis or acute symptoms are present [10]. Furthermore, anatomic variations such as *dens invaginatus* or *dens in dente* are associated with complex internal anatomy, which also renders the access cavity difficult [11,12,13,14].

It is well-known that access cavities should be kept as conservative as possible, following the trend of a minimally invasive dentistry. The achieved tooth preservation may be an efficient way to decrease the occurrence of post-treatment tooth fractures [15]. Access cavity preparations involving both mesial and distal marginal ridges can reduce cuspal stiffness by up to 63% [16]. Although nomenclature in science is crucial to convey ideas and concepts unambiguously, the standardization of access cavity terminology is still an issue. Generally, whether streaming from social media or purely scientific lines, approaches such as “ninja” and truss cavities that preserve the dentinal bridge by producing two or more occlusal access multi-rooted teeth have flourished. The most widely accepted concepts for access cavities are the traditional, the conservative with parallel or divergent walls, the ultra-conservative, the caries-driven, the restorative-driven and the truss cavity [15]. In all the above-mentioned clinical scenarios, preparing an adequate access cavity and identifying root canal(s) orifice(s) can be demanding and may result in an excessive loss of tooth structure, ultimately leading to increased fracture susceptibility. Therefore, preoperative planning is highly recommended and three-dimensional (3D) imaging may be a helpful tool.

Cone-beam computed tomography (CBCT) is considered a valuable tool for the diagnosis and treatment planning in endodontics, frequently helping clinicians in establishing a proper PCO approach [17,18,19,20]. CBCT presents a noninvasive imaging and measuring technology often used in oral implantology for 3D planning, anatomic structures visualization and bone level quantification, as well as guided implant surgery using printed templates [1]. CBCT should not be routinely used, solely being considered when meticulous clinical and two-dimensional radiographic examinations are inconclusive to clarify complex situations per ALARA principles (“as low as reasonably achievable”) [21].

Guided endodontics, either with computer-aided static (SN) or dynamic navigation (DN) techniques, has recently emerged as an alternative for access cavity preparation in the clinical management of complex cases [22], including partial or total root canal space calcifications, anomalous teeth, glass fiber post removal, *dens invaginatus* and *evaginatus,* dystrophic calcification, teeth presenting with rotations and/or versions, teeth with fixed prosthetic rehabilitation, selective root canal treatments, access through calcium silicate-based cements (e.g., MTA), non-surgical endodontic retreatment and microsurgical endodontic procedures [9,23,24,25,26,27,28,29,30,31,32].

Computer-aided SN and DN techniques are based on CBCT datasets. Static guidance refers to the utilization of fixed surgical guides, which are produced using computer-aided design and computer-assisted manufacturing (CAD/CAM) [25,33]. On the other hand, DN requires an optical triangulation tracking system that uses real-time stereoscopic motion-tracking cameras to guide the drilling process according to the planned angle, pathway, and depth of endodontic access cavities [2].

These modern techniques have improved the predictability of endodontic access cavities preparation and root canal location [34,35,36], with several studies reporting higher volume of dental tissue preservation with the digitally-guided over conventional freehand access [5,37,38,39,40,41,42].

The present review aims to compare SN and DN guided endodontics’ techniques in non-surgical endodontic treatment.

## 2. Materials and Methods

An electronic search was carried out using the following databases: PubMed, Scopus and Cochrane Library. The search assessed all literature published until October 2021. In Pubmed database, the search terms were applied as follows: (((endodontics) OR ((endodontics) [MeSH Terms])) AND guided). In both Cochrane Library and Scopus databases, the following search key was used: (guided AND endodontics). Search was restricted by language (English, Spanish and Portuguese) and year of publication (from 2011 to 2021).

The inclusion criteria encompassed all types of study designs addressing endodontic access cavities achieved with computer-aided technology (SN or DN techniques) in non-surgical endodontic treatment.

The exclusion criteria included the use of computer-aided technology in endodontic microsurgery, implantology, bone regeneration, periodontology, laser therapy and restorative dentistry. Other articles non-related to the clinical application of SN or DN techniques in non-surgical endodontic treatment and studies in which CBCT was used only as a diagnostic method were also excluded.

After duplicates deletion, the titles and abstracts were screened based on the inclusion and exclusion criteria. The selected studies were subsequently subjected to a full-text evaluation for eligibility assessment.

## 3. Results

After the initial search, a total of 449, 168 and 32 articles were identified in PubMed, Scopus, and Cochrane Library databases, respectively. Of the 649 articles, 134 duplicates were discarded. In this case, 67 articles were selected after title and abstract screening, of which 60 were assessed for eligibility through full-text analysis, with one article being excluded. Lastly, 4 cross-references were added. Thus, a total of 63 studies were included (Figure 1). Retrieved data from the analyzed studies are described for original research studies (Table 1) and case reports/series (Table 2).

In this case, 14 original research studies focused on computer-aided static technique and 9 on the dynamic navigation method. Only one study comparatively evaluated both guided endodontics’ techniques. Moreover, 24 case reports and 6 case series assessed static technique’s clinical and/or radiographic outcomes.

## 4. Discussion

An adequate access cavity preparation presents a key step for the success of non-surgical endodontic treatment, with excessive substance loss affecting the long-term prognosis of the tooth [44].

Numerous alterations–calcified canals and several anatomic variations–render non-surgical endodontic treatment more complex. In order to overcome these difficulties and optimize the outcome of challenging endodontic cases, 3D printed guides were introduced based on principles similar to those of guided implant surgery. In the endodontic field, 3D templates are used for root canal location during non-surgical endodontic treatment, mainly when there are significant risks of procedural errors, including root perforation, which can severely jeopardize treatment outcomes [69]. Connert et al. [46] compared conventional freehand access cavity preparation with the static guided approach using 3D printed teeth with simulated calcified root canals, ultimately showing that access cavities resulting from guided endodontics application allowed maximum tooth substance conservation. In fact, access cavities obtained with guided endodontics were comparable with the recently reported minimally invasive access (MIA) cavities. Additionally, guided endodontics was associated with a significantly lower procedure time [37,51,52]. In addition, while 91.7% of the root canals were located and negotiated using guided endodontics, only 41.7% of the root canals were accessed and negotiated when applying the conventional access preparation technique. In conclusion, these results confirm that the guided technique outperforms the conventional approach regarding root canal location and intervention time [46]. Furthermore, although operator experience seems to play a key role when accessing severely calcified teeth with conventional freehand access, computer-aided techniques have been reported as operator-independent [1,26,44,48,51]. Therefore, digital techniques would be especially valuable for the management of complex cases by less experienced operators.

The workflow of SN technique demands a cone-beam computed tomographic (CBCT) and intraoral scanning to produce a template used to guide the bur through an ideal path previously defined based on the information provided by both techniques [48]. Firstly, a high-resolution preoperative CBCT scan is taken and stored in Digital Imaging and Communication (DICOM) format [44,56,57,65]. Afterward, a digital intraoral impression of the patient’s upper and lower arch is obtained using an intraoral scanner [57,70]. This step can also be indirectly obtained from gypsum cast [56,68]. The resulting file must be saved in surface tessellation language (STL). The two files are then uploaded to a specialized image processing software [58,70]. The following step is to design a 3D template using 3D design software, where the bur and the corresponding sleeve positions are planned [44,45,54,56,59,70]. Finally, the 3D guide needs to be printed or milled, and the metal sleeves are incorporated into the template [44,45,56,60,70]. Before the endodontic treatment, the teeth that will support the guide should be isolated with rubber dam and then the guide’s fitting and stability must be checked [59,61]. With the 3D template correctly set, the canal can be prepared up to the established working length by inserting the bur through the metal sleeve [34,59]. Once the root canal system is located, it can be negotiated with endodontic hand files until the working length and copiously irrigated. After this, the endodontic treatment can be performed based on conventional root canal preparation and filling techniques [59,61].

DN uses a mobile unit which includes an overhead light, a stereoscopic motion-tracking camera, and a computer with implant planning software. These items are used to guide in real time a calibrated handpiece until reaching the reference point previously determined [49]. To perform access cavities through DN, likewise the SN approach, it is first necessary to take a high-resolution preoperative CBCT scan. In order to digitally plan the entry point of the bur, its pathway, depth and angle, the CBCT scan file is uploaded to the dynamic navigation system (DNS) planning software [51]. Before opening the access cavity, the tooth or teeth should be isolated with rubber dam [50]. An initial calibration process of the handpiece with the jaw according to the producer’s indications is necessary. The access cavity can be subsequently performed with simultaneous monitoring by the DNS [51,53]. The operator can visually control the progression of the bur by watching it on the laptop screen. In the real-time image, the depth of the bur is indicated by a green bar on the depth gauge; when within 1 mm of the desired depth, the bar’s color changes from green to yellow and thereafter to red, when the planned depth is reached [50]. Similarly, to the static approach, once the root canal system is located, it can be negotiated [61].

Several studies confirm the accuracy of both static [1,5,26,44] and dynamic [10,16,47,49,50] approaches. According to a previous systematic review and meta-analysis, there are no statistically significant differences between both computer-aided approaches in terms of root canal location rate, with SN and DN exhibiting success rates of 98.5% and 94.5%, respectively [2]. Moreover, suitability of the static technique for predictable and accurate fiber post removal has also been disclosed [27].

Both static and dynamic systems present correspondent advantages and disadvantages. A major advantage of digital planning is that it is possible to preoperatively visualize the root canal location and plot the navigation in detail without having to mentally transfer the planning to the clinical situation. This will allow dentists to achieve predictable results without extensive endodontic skills being required [61]. One crucial drawback to consider is that the PCO extension might affect the accuracy of both computer-aided techniques [2]. Regardless of the technique, the quality of the CBCT presents a crucial aspect for a correct preoperative planning. Spatial resolution of the CBCT does not always allow for the visualization of the root canal and the presence of highly radiopaque materials can result in radiographic artifacts which jeopardize digital planning [52,53].

The SN technique performed with surgical templates avoids the need for drilling guidance during treatment [53]. It was also mentioned that SN reduced excessive loss of tooth structure and chair-side operating times. Furthermore, using a single bur or two burs ensured the accuracy of the drilling procedure [69]. Despite the advantages of this method, the approach exhibits several disadvantages, one of them being that multiple teeth must be isolated during the procedure because the guide must fit directly on the teeth to ensure its stability [61,62]. Moreover, as the guide restricts visual access to the endodontic cavity, its removal is required whenever the operator aims to confirm the path during the treatment [61,67]. Specifically in posterior teeth, static guidance requires the fabrication of several templates to allow straight access to individual root canals. In addition, it is noteworthy that the accuracy of the technique greatly relies on the surgical template’s design and manufacturing process [53]. Regarding clinical application limitations, the static-guided approach requires a straight path to the target, therefore being difficult to use this technique in cases of small or limited mouth openings and/or in posterior teeth with reduced interocclusal space. However, an intracoronal guide has been proposed to overcome this difficulty. In addition, larger diameter slow-speed drills can generate cracks on the tooth surface and produce excessive heat that may harm the periodontal ligament. Finally, the lack of 3D real-time visualization prevents intraoperative changes in the predefined drill trajectory [25,50,52,61,63,64,66].

One of the main benefits of DN is that it enables a direct view of the operatory field and allows clinicians to readjust the direction of the endodontic access cavity bur in real-time [53]. In addition, there is no need for an intraoral scan, the overheating risk deriving from the lack of barrier between the water source and the bur is reduced, and it is especially useful in cases of limited mouth opening or posterior region treatments since a template is not necessary [51]. Additionally, endodontic urgencies can be treated using DN, whereas the static approach requires an additional step of template design and printing [47,51]. Despite the accuracy of DN, disadvantages and limitations were also outlined. The primary disadvantages are the difficulty of keeping the system display in sight during the procedure and the long learning curve of the technique, with proper operator training prior to treatment being required [2,63]. Nevertheless, it has been suggested that augmented reality technology can be employed to transmit the virtual image displayed by the DNS while maintaining the therapeutic area’s visibility [2]. Moreover, high initial equipment investment is involved [47].

Overall, guided endodontics allows for obtaining MIA and predictably locating root canals in complex cases while minimizing the potential risk of iatrogenic damage and enabling greater tooth structure preservation [25,39,46], thus presenting a more accurate and safe treatment option over conventional freehand techniques [53]. Since the SN technique was introduced earlier, it is better supported by scientific literature than the dynamic method [2]. The latter, however, shows higher potential for future developments and improvements. Nevertheless, further studies with larger sample size and standardized protocols are currently necessary to consolidate the precision of both guided endodontics’ methods [38]. Moreover, research focusing on different clinical scenarios would be of uttermost importance. Although numerous case reports and few case series are currently available, randomized clinical trials would provide valuable data on the clinical outcome of guided endodontics application.

## 5. Conclusions

The clinical management of complex endodontic cases, such as severely calcified canals, can be challenging for clinicians. Guided endodontics procedures present minimally invasive and accurate techniques which allow for highly predictable root canal location, greater tooth structure preservation and lower risk of iatrogenic damage, mainly when performed by less experienced operators. Both SN and DN approaches exhibit different advantages and disadvantages that make them useful in distinct clinical scenarios. Therefore, it is important to rigorously evaluate each clinical case for subsequent adequate guided endodontics technique selection.

## Figures and Tables

**Figure 1 jpm-12-01516-f001:**
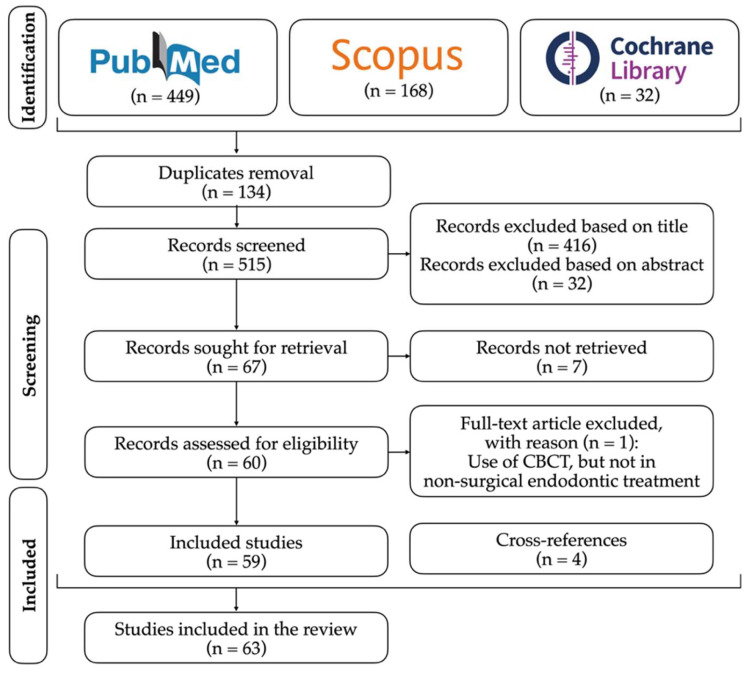
Flowchart of studies selection.

**Table 1 jpm-12-01516-t001:** Included original research studies.

Author (Year)	Study Type	Aim	Sample	Included Groups	Computer-Aided Technique	Results	Main Conclusions
Krug et al. (2020) [5]	In vitro	To compare the accuracy and effort of digital workflow for GEA procedures using two different software applications in 3D-printed teeth modeled to simulate PCO in vitro.	32 3D-printed teeth with simulated PCO	Incisors	SN	**Root canal location rate:** SE–100%; CDX–93.8% **Angle deviation** (*p* < 0.001): CDX-1.57 (1.16–1.97)° SE–0.68 (0.47–0.90)° **Labial-oral deviation** (*p* < 0.001): CDX–0.54 (0.37–0.71) mm SE–0.12 (0.06–0.18) mm **Mesiodistal and Coronal-apical deviation** (*p* > 0.05)**:** no statistically significant difference **3D vector deviation** (*p* < 0.001): CDX–0.74 (0.60–0.87) mm SE–0.35 (0.26–0.43) mm **Planning time** (*p* < 0.05): CDX–10 min 50 s (4 min 16 s–17 min 24 s) SE–20 min 28 s (11 min 2 s–29 min 54 s) **Number of computer clicks** (*p* < 0.01): CDX–107 (62–151) SE–341 (208–473)	Both methods enabled rapid drill path planning, a predictable GEA procedure, and the reliable location of root canals in teeth with PCO without perforation.
Leontiev et al. (2021) [22]	In vitro	To demonstrate that MRI is sufficiently accurate for the detection of root canals using guided endodontics.	100 human teeth	Anterior and premolar	SN	**Root canal location rate:** 91% **Angle deviation:** 1.82 (0.00–7.60)° **Bur‘s base deviation:** 0.21–0.31 mm **Bur‘s tip deviation:** 0.28–0.44 mm Preparation in the buccolingual dimension was significantly more precise in mandibular compared with maxillary teeth, and accuracy in the mesiodistal dimension was more precise in anterior teeth compared with premolars.	This in vitro study demonstrated the suitability of MRI for guided endodontic access cavity preparation.
Perez et al. (2021) [27]	In vitro	To evaluate accuracy of surgically guided endodontics when applied to the endodontic retreatment of canals with fibre-posts.	40 human teeth	Molars and bicuspids	SN	**Apical gutta-percha location rate**: 87.5% **Coronal stage deviation** (n = 40): Buccal-oral–0.23 ± 0.18 mm Mesial-distal–0.28 ± 0.12 mm Global–0.39 ± 0.14 mm **Apical stage deviation** (n = 35): Buccal-oral–0.24 ± 0.20 mm Mesial-distal–0.26 ± 0.15 mm Global–0.40 ± 0.19 mm **Difference between operators:** Not found	Microguided endodontics is a predictable and accurate method to remove fibre-post restorations efficiently.
Nayak et al. (2018) [43]	In vitro	Not mentioned	6 human teeth	Single-rooted human teeth (incisors, laterals, canines, and premolars)	SN	**Maximum deviation found:** 0.210 ± 0.04 mm ∆X = 0.134 ± 0.03 mm ∆Y = 0.145 ± 0.05 mm ∆D = 0.210 ± 0.04 mm ∆E = 0.07 ± 0.02 mm	The proposed method reveals encouraging results for endodontic guide design.
Conner et al. (2017) [44]	In vitro	To assess the accuracy of guided endodontics in mandibular anterior teeth by using miniaturized instruments.	60 human teeth	Mandibular incisors and canines	SN	**Total time required:** 613 (447–936) s **Deviation Angle:** 1.59 (0–5.3)° **Bur’s base deviation:** Mesial-distal–0.12 (0–0.54) mm Buccal-oral–0.13 (0–0.4) mm Apical-coronal–0.12 (0–0.41) mm **Bur’s tip deviation:** Mesial-distal–0.14 (0–0.99) mm Buccal-oral–0.34 (0–1.26) mm Apical-coronal–0.12 (0–0.4) mm **Difference between operators:** Not found	Microguided endodontics provides an accurate, fast, and operator-independent technique for the preparation of apically extended access cavities in teeth with narrow roots such as mandibular incisors.
Zehnder et al. (2016) [1]	Ex vivo	To present a novel method utilizing 3D printed templates to gain guided access to root canals and to evaluate its accuracy in vitro.	60 human teeth	Incisors, laterals, canines and premolars (single-rooted human teeth)	SN	**Root canal location rate:** 100% **Angle deviation:** 1.81 (0–5.6)° **Bur‘s base deviation:** Apical/coronal–0.16 (0–0.76) mm Mesial/distal–0.21 (0–0.75) mm Buccal/oral–0.2 (0–0.76) mm **Means of deviations at bur’s base:** 0.16–0.21 mm **Bur‘s tip deviation:** Apical/coronal–0.17 (0–0.75) mm Mesial/distal–0.29 (0–1.34) mm Buccal/oral–0.47 (0–1.59) mm **Means of deviations at bur’s tip:** 0.17–0.47 mm **Difference between operators:** Not found	‘Guided Endodontics’ allowed an accurate access cavity preparation up to the apical third of the root utilizing printed templates for guidance. All root canals were accessible after preparation.
Buchgreitz et al. (2016) [35]	Ex vivo	To evaluate ex vivo, the accuracy of a preparation procedure planned for teeth with PCO using a guide rail concept based on a CBCT scan merged with an optical surface scan.	48 human teeth	Not mentioned	SN	**Root canal location rate:** 79.2% **Apical horizontal deviation** (*p* < 0.001): 0.46 (0.69–0.32) mm	The combined use of CBCT and optical scans for the precise construction of a guide rail led to a drill path with a precision below a risk threshold. The present technique may be a valuable tool for the negotiation of partial or complete pulp canal obliteration.
Krug et al. (2020) [45]	In vitro	To compare the accuracy and effort of digital workflow for GEA procedures using two different software applications in 3D-printed teeth modeled to simulate PCO in vitro.	32 3D-printed teeth with simulated PCO	Incisors	SN	**Root canal location rate:** SE–100% CDX–93.8% **Angle deviation** (*p* < 0.001): CDX–1.57 (1.16–1.97)° SE–0.68 (0.47–0.90)° **Labial-oral deviation** (*p* < 0.001): CDX–0.54 (0.37–0.71) mm SE–0.12 (0.06–0.18) mm **Mesiodistal and Coronal-apical deviation** (*p* > 0.05)**:** no statistically significant difference **3D vector deviation** (*p* < 0.001): CDX–0.74 (0.60–0.87) mm SE–0.35 (0.26–0.43) mm **Planning time** (*p* < 0.05): CDX–10 min 50 s (4 min 16 s–17 min 24 s) SE–20 min 28 s (11 min 2 s–29 min 54 s) **Number of computer clicks** (*p* < 0.01): CDX–107 (62–151) SE–341 (208–473)	Both methods enabled rapid drill path planning, a predictable GEA procedure, and the reliable location of root canals in teeth with PCO without perforation.
Ali et al. (2021) [25]	In vitro	To evaluate the effectiveness of the SN technique for accessing the root canal through the mineral trioxide aggregate (MTA) and to evaluate the effect of this technique on the fracture strength of teeth.	30 human teeth	Mandibular premolars	SN versus FH	**Root canal location rate**: 100% **Number of cases with mishaps** (*p* < 0.05): FH–13 SN–0 **Root canals penetration time** (*p* < 0.05): SN < FH **Fracture strength** (*p* < 0.05): FH–472.3 Newtons SN–1379.8 Newtons **Non-restorable failure** (*p* < 0.05): FH–13 SN–3	Within the limitations of the present study, the SN technique yielded favorable results with respect to time, mishaps, and fracture strength.
Choi et al. (2021) [37]	In vitro	To determine the effectiveness of using an access opening guide in teaching ideal access opening shape and preventing excessive tooth loss, with a focus on predoctoral dental students.	90 human teeth	60 premolars and 30 molars	SN versus FH	**Access opening times–premolar group**: FH–327.2 ± 135.5 s SN–97.4 ± 106.6 s **Access opening times–molar group**: FH–547.43 ± 269.6 s SN–104.57 ± 55.5 s **Volume differences–premolar group**: FH–38.1 ± 32.2 mm^3^ SN–−2.0 ± 14.4 mm^3^ **Volume differences–molar group**: FH–72.2 ± 60.6 mm^3^ SN–−8.7 ± 16.8 mm^3^	Using the AOG-3DP (3D printer) significantly reduced the access opening time for premolars and molars. However, there is a limitation in that CBCT DICOM images must be converted to stereolithographic. stl files in order to be printed via 3D technology. This requires additional preclinical treatment time for imaging and subsequent printing. It could be considered that this can be a useful method in difficult cases.
Kostunov et al. (2021) [39]	In vitro	To investigate how much tooth substance can be preserved by using the guided approach.	30 acrylic typodont teeth	Incisors (10), premolars (10), molars (10),	SN versus FH	**Root canal location rate**: SN–93.3% FH–100% **Tooth substance removal (FH)**: Incisors–16.1 ± 3.7 mm^3^ Premolars–44.2 ± 8.9 mm^3^ Molars–99.3 ± 3.1 mm^3^ **Tooth substance removal (SN)**: Incisors–10.3 ± 1.1 mm^3^ Premolars–29.3 ± 4.2 mm^3^ Molars–51.8 ± 5.3 mm^3^ mm^3^ **Tooth substance removal (FH vs. SN)**: *p* < 0.05	Use of guided endodontics in normally calcified teeth enables preservation of a significant amount of tooth substance. This advantage must be carefully balanced against a greater radiation burden and risk of perforation, higher costs, and more difficult debridement and visualization of the pulp chamber and root canals.
Connert et al. (2019) [46]	In vitro	To compare endodontic access cavities in teeth with calcified root canals prepared with the conventional technique and a guided endodontics approach regarding the detection of root canals, substance loss, and treatment duration.	48 3D-printed teeth with simulated calcified root canals	Upper and lower incisors	SN versus FH	**Root canal location rate:** FH–41.7% SN–91.7% **Substance loss:** FH–49.9 (42.2–57.6) mm^3^ SN–9.8 (6.8–12.9) mm^3^ **Treatment duration:** FH–21.8 (15.9–27.7 min) SN–11.3 (6.7–15.9 min) **Operator experience:** does not influenced the success of the guided approach	Guided endodontics allows a more predictable and expeditious location and negotiation of calcified root canals with significantly less substance loss.
Loureiro et al. (2020) [38]	Ex vivo	To compare the volume of dental tissue removed after GEA and CEA to mandibular incisors and upper molars.	40 human teeth	20 mandibular incisors (G1) and 20 upper molars (G2)	SN versus FH	**Volume of dental tissue removed (*p* = 0.004):** G1 group using CEA–31.667 mm^3^ (10.62%) G1 group using GEA–26.523 mm^3^ (10.65%) G2 group using CEA–62.526 mm^3^ (5.86%) G2 group using GEA–45.677 mm^3^ (4.11%)	GEA preserved a greater volume of dental tissue in extracted upper human molars than CEA; however, there was no significant difference between CEA and GEA in the volume of dental tissue removed from mandibular incisors.
Wang et al. (2021) [40]	In vitro	To evaluate in a laboratory setting, the impact of three designs of endodontic access cavities on dentine removal and effectiveness of canal instrumentation in extracted maxillary first molars using micro-CT.	30 human teeth	Maxillary first molars	SN (GEC) versus TEC versus CEC	**Total volume of dentin removed:** TEC–81.78 ± 14.77 mm^3^ CEC–39.26 ± 8.99 mm^3 ^ GEC–42.39 ± 8.17 mm^3^ TEC group > CEC and GEC group (*p* < 0.05). The volume of dentine removed in the crown, pericervical dentine and coronal third of the canal was significantly lower in CEC and GEC groups when compared to that in the TEC group (*p* < 0.05). There was no significant difference in noninstrumented canal area, canal transportation and centring ratio amongst the TEC, CEC and GEC groups (*p* > 0.05).	CEC and GEC preserved more tooth tissue in the crown, pericervical dentine and coronal third of the canal compared with TEC after root canal preparation. The design of the endodontic access cavity did not impact on the effectiveness of canal instrumentation in terms of noninstrumented canal area, canal transportation and centring ratio.
Torres et al. (2021) [47]	In vitro	To evaluate 3D accuracy and outcome of a dynamic navigation method for guided root canal treatment of severe PCO in 3D printed jaws.	132 3D printed teeth with PCO	Anterior teeth, premolars, and molars	DN	**Root canal location rate:** 92.9% **Mean deviation at the apical point:** 0.63 mm ± 0.35 *p* < 0.05: anterior teeth < molars **Mean angular deviation:** 2.81° ± 1.53	DN was an accurate approach for root canal treatment in teeth with severely calcified canals. However, the technique has a learning curve and requires extensive training prior to its use clinically.
Chong et al. (2019) [16]	In vitro	To investigate the use of DN for guided endodontics.	29 human teeth	Incisors, laterals, canines, premolars and molars	DN	**Root canal location rate:** 89.1%	The results of this study demonstrate the potential of using DN technology in guided endodontics in clinical practice.
Jain et al. (2020) [10]	In vitro	To present a novel dynamic navigation method to attain minimally invasive access cavity preparations and to evaluate its 3D accuracy in locating highly difficult simulated calcified canals among maxillary and mandibular teeth.	84 printed teeth with simulated PCO	Anterior, premolars and molars	DN	**2D horizontal deviation–canal orifice**: 0.9 ± 0.69 mm (*p* < 0.05): Maxilla: 1.0 ± 0.78 mm Mandible: 1.0 ± 0.60 mm **3D deviation–canal orifice**: 1.3 ± 0.65 mm (*p* ≥ 0.05): Maxilla: 1.2 ± 0.57 mm Mandible: 1.4 ± 0.70 mm **3D angular deviation**: 1.7 ± 0.98° (*p* < 0.05): Premolars: 1.4 ± 0.62° Molars: 1.9 ± 1.14° The 3D and 2D discrepancies were independent of the canal orifice depths (*p* > 0.05). The average drilling time was 57.8 s with significant dependence on the canal orifice depth, tooth type, and jaw (*p* < 0.05)	This study shows the potential of applying DN technology with high-speed drills to preserve tooth structure and accurately locate root canals in teeth with pulp canal obliteration.
Pirani et al. (2020) [19]	In vitro	To evaluate the potential application of dynamic navigation in teaching undergraduate students the opening of the access cavity.	3 human teeth	2 lower molars and 1 lower premolar	DN	**Root canal location rate**: 100%	Present results demonstrated a possible application of this technology for educational purposes in finding root canals.
Connert et al. (2021) [48]	In vitro	To evaluate substance loss and the time required for ACP using the FH method versus a DN system of real-time guided endodontics in an in vitro model using 3D printed teeth.	72 3D-printed teeth	Maxillary central incisors, lateral incisors, and canines	DN versus FH	**Root canal location rate:** 97.2% **Substance loss** (*p* < 0.001): FH–29.7 (24.2–35.2) mm^3^ DN–10.5 (7.6–13.3) mm^3^ **Time required for ACP** (*p* > 0.05): FH–193 (164–222) s DN–195 (135–254) s **Operator experience:** More experienced operator removed significantly (*p* < 0.001) less tooth structure (19.9 mm^3^) than the second operator (39.4 mm^3^) when performing FH.	DN is a practicable, substance-sparing method performed in comparable time as FH. Moreover, DN seems to be independent of operator experience.
Gambarini et al. (2020) [49]	In vitro	To evaluate the possible use of a novel DN in planning and executing ultra conservative access cavities, and its precision in vitro, compared to a FH approach without any guide.	20 artificial teeth replicas	Upper right first molars	DN versus FH	**Root canal location rate**: 100% for FH and DN groups **Deviation Angle** (*p* < 0.05): FH–19.2 ± 8.6° DN–4.8 ± 1.8° **Maximum distance from the ideal position** (*p* < 0.05): FH–0.88 ± 0.41 mm DN–0.34 ± 0.19 mm **Instrumentation time** (*p* > 0.05): FH–12.2 ± 3.2 s DN–11.5 ± 2.4 s	The use of DN increased the benefits of ultra-conservative access cavities, by minimizing the potential risk of iatrogenic weakening of critical portions of the crown and reducing negative influences to shaping procedures.
Jain et al. (2020) [50]	In vitro	To compare the speed, qualitative precision, and quantitative loss of tooth structure with FH and DN access preparation techniques for root canal location in 3D–printed teeth with simulated calcified root canals.	40 3D- printed teeth to simulate canal calcification	Maxillary (21) and mandibular (41) single-rooted central incisors	DN versus FH	**Quantitative Substance Loss** (*p* = 0.0356): FH–40.7 (29.1–52.2) mm^3^ DN–27.2 (22.0–32.5) mm^3^ **Qualitative Precision (optimal)** (*p* > 0.05): FH–45% DN–75% **Qualitative Precision (suboptimal)**: FH–40% DN–15% **Qualitative Precision (unacceptable)**: FH–15% DN–10% **Treatment duration** (*p* < 0.05): FH–424.8 (289.4–560.2) s DN–136.1 (101.4–170.8) s	Overall, DN access preparations led to significantly less substance loss with optimal and efficient precision in locating simulated anterior calcified root canals in comparison with freehand access preparations.
Dianat et al. (2020) [51]	Ex vivo	To compare the accuracy and efficiency of a DN system to the FH method for locating calcified canals in human teeth.	60 human teeth with PCO	Single-rooted teeth (maxillary and mandibular incisors, canines, and premolars)	DN versus FH	**Root canal location rate** (*p* > 0.05)**:** FH–83.3% DN–96.7% **Linear deviation (BL;** *p* ≤ 0.001)**:** FH–0.81 ± 0.74 mm DN–0.19 ± 0.21 mm **Linear deviation (MD;** *p* > 0.05)**:** FH–0.31 ± 0.35 mm DN–0.12 ± 0.14 mm **Angular deflection** (*p* ≤ 0.0001)**:** FH–7.25 ± 4.2° DN–2.39 ± 0.85° **Reduced Dentin Thickness (CEJ;** *p* ≤ 0.0001)**:** FH–1.55 ± 0.55 mm DN–1.06 ± 0.18 mm **Reduced Dentin Thickness (End drilling point;** *p* ≤ 0.001)**:** FH–1.47 ± 0.49 mm DN–1.18 ± 0.17 mm **Time required for ACP (***p* ≤ 0.05) (s) FH–405 ± 246 s DN–227 ± 97 s **Mishaps (***p* ≤ 0.05)**:** FH–8 DN–1 **Frequency of Successful Attempts** (*p* > 0.05): FH–25/30 DN–29/30 **Operator experience:** The time required for ACP was significantly shorter for the board-certified endodontist in the FH group (*p* ≤ 0.05). The rest of the measured variables were not different between the 2 operators (*p* > 0.05)	The DN system was more accurate and more efficient than the FH technique in locating calcified canals in human teeth. This novel DN system can help clinicians avoid catastrophic mishaps during access preparation in calcified teeth.
Janabi et al. (2021) [52]	In vitro	To investigate the accuracy and efficiency of the 3D DN system compared with the FH technique when removing fiber posts from root canal–treated teeth.	26 human teeth	Maxillary single-rooted teeth (maxillary canines and incisors)	DN versus FH	**Coronal deviation** (*p* < 0.05): **FH**–1.13 ± 0.84 mm **DN**–0.91 ± 0.65 mm **Apical deviation deviation** (*p* < 0.05): **FH**–1.68 ± 0.85 mm **DN**–1.17 ± 0.64 mm **Angular deflection** (*p* < 0.05): **FH**–4.49 ± 2.10° **DN**–1.75 ± 0.63° **Operation time** (*p* < 0.05): **FH**–8.30 ± 4.65 min **DN**–4.30 ± 0.43 mi **Volumetric loss of tooth structure** (*p* < 0.05): DN < FH	The DN system was more accurate and efficient in removing fiber posts from root canal–treated teeth than the FH technique.
Zubizarreta-Macho et al. (2020) [53]	In vitro	To analyze the accuracy of two computer-aided navigation techniques to guide the performance of endodontic access cavities compared with the FH procedure.	30 human teeth	Single-rooted anterior teeth (lower central incisors))	SN versus DN versus FH	**Coronal deviation (*p***-**value):** SN-DN = 0.654 SN-FH < 0.001 DN-FH < 0.001 **Apical deviation (*p***-**value):** SN-DN = 0.914 SN-FH < 0.001 DN-FH < 0.001 **Angular deviation (*p***-**value):** SN-DN = 0.072 SN-FH < 0.001 DN-FH < 0.001	SN and DN procedures allow more accurate and safe endodontic access to cavities than conventional freehand techniques.

2D: 2 dimensional; 3D: 3 dimensional; ACP: access cavity preparation; CBCT: cone-beam computed tomography; CDX: CoDiagnostiX; CEA: conventional endodontic access; CEC: conservative endodontic cavity; DN: computer-aided dynamic navigation; FH: freehand conventional technique; GEA: guided endodontic access; GEC: guided endodontic cavity; MRI: magnetic resonance imaging; PCO: pulp canal obliteration; SE: Sicat Endo; SN: computer-aided static navigation; TEC: traditional endodontic cavity.

**Table 2 jpm-12-01516-t002:** Included case reports/series.

Author (Year)	Study Type	Aim	Tooth	Computer-Aided Technique	Outcome		Main Conclusions
					Follow-up	Clinical and/or Radiographic	
Casadei et al. (2020) [3]	Case Report	To describe endodontic treatment where there was an intercurrence, generating deviation and perforation, which was solved with the aid of guided endodontics.	Calcified maxillary right second premolar	SN	1 year	**Clinical:** Absence of painful symptoms and a negative response to clinical tests. **Radiographic:** Regression of the periapical lesion.	This technique proved to be safe and predictable, allowing for a favorable prognosis in the long term.
Zubizarreta-Macho et al. (2019) [11]	Case Report	To propose a novel technique, based on new technologies, to make access cavity conservative and guided with minimal dental structure lost.	Retreatment of a DI in a maxillary left lateral incisor	SN	6, 12 and 18 months	**Clinical:** Absence of clinical signs. **Radiographic:** Reduction of the periapical lesion.	The splint guide allowed a guided and conservative access cavity to root canal system. It facilitates the root canal retreatment and improves the prognosis of the teeth with dental malformations.
Zubizarreta-Macho et al. (2015) [12]	Case Report	To describe the treatment of a type II DI by means of guided splints for cavity access.	Maxillary left lateral incisor with type II DI	SN	6 months	**Clinical:** Any signs or symptoms. **Radiographic:** Periapical lesion disappearance.	CBCT is an effective method for obtaining information about the root canal system in teeth with DI. Guided implant surgery software is effective for manufacturing splint guides for endodontic treatment with conservative pulp chamber access.
Ali et al. (2019) [13]	Case Report	To describe the use of the guided endodontics technique for two maxillary lateral incisors with multiple DI.	Maxillary lateral incisors with type II DI	SN	Not mentioned	**Radiographic:** Regression of the periapical lesion in relation to teeth 21 and 22.	This technique can be a valuable tool for negotiation of the DI and the main root canal, reducing chair-time and, more importantly, the risk of iatrogenic damage to the tooth structure.
Ishak et al. (2020) [20]	Case Report	To evaluate the benefits and limitations of the micro-guided technique.	Calcified central lower incisors	SN	Not mentioned	Not mentioned	The guided approach allows predictable, efficient endodontic treatment of teeth presenting calcified canals, with minimal removal of sound dentine and less risk of root perforations.
Maia et al. (2019) [54]	Case Report	To describe a protocol for adhesive fiber post removal using a prototyped endodontic guide.	Maxillary right central incisor restored with a fiber post	SN	Not mentioned	Not mentioned	The CAD/CAM technology to generate guides with prototyping is a useful tool for fiber post removal.
Perez et al. (2020) [27]	Case Report	To demonstrate the usefulness of endodontic guides for the removal of fiber posts.	Maxillary right first molar restored with a fiber post	SN	12 months	**Radiographic:** Apical area healed.	This case study illustrates the benefits of endodontic guides for the removal of fibre posts. It also demonstrates the feasibility of the technique in the posterior segment where the bulkiness of the device might appear to be a drawback.
Mena-Álvarez et al. (2017) [55]	Case Report	To describe the treatment of a type V *dens evaginatus* by using splits as guides to perform access cavity.	Maxillary left central incisor with type V *dens evaginatus*	SN	1 year	**Radiographic:** Recovery and disappearance of the periapical lesion.	CBCT is an effective method for obtaining internal anatomical information of teeth with anatomical malformations. The osseointegrated implant planning software is an effective method for planning root canal treatment and designing stereolithograped splits (for performing minimally invasive access cavities)
Maia et al. (2021) [28]	Case Report	To describe the use of a prototyped guide created with virtual planning for fiber-reinforced composite resin post removal.	Maxillary left central incisor restored with a fiber post	SN	18 months	**Radiograpchic:** Apical lesion healed.	The prototyped guide improved patient safety, decreased professional stress, shortened the treatment time, preserved the esthetics, and eliminated the need for a new restoration. In addition, this approach does not require specialized training or extensive clinical experience to achieve predictable results.
Maia et al. (2020) [28]	Case Report	To describe the use of the guided endodontics for a non-surgical endodontic retreatment of the mandibular molar.	Nonsurgical re-treatmen of a mandibular right first molar	SN	24 months	**Radiographic:** Reduction of the periapical lesion, suggestive of complete healing.	The EndoGuide technique can be indicated as a fast, available and accurate solution for endodontic therapy in calcified root canals.
Santiago et al. (2022) [31]	Case Report	To describe the CAD/CAM workfow to create personalized templates with innovative design and the guided endodontics’ clinical procedures to treat the obliterated mesiobuccal and mesiolingual root canals of a mandibular molar. To compare the open-template design to the compact design and report the success assessment after one year of follow-up.	Mandibular right first molar with dystrophic calcifcation	SN	1 year	**Clinical:** Asymptomatic tooth, negative to percussion tests. **Radiographic:** Integrity of the adjacent tissues.	The digital planning and guided access permitted to overcome the case limitations and then re-establish the glide path following the original anatomy of the root canals. The guided endodontic represents a personalized technique that provides security, reduced risks of root perforation, and a significant decrease of the working time to access obliterated root canals even in the mesial root canal of mandibular molars, a region of limited mouth opening.
Kaur et al. (2021) [56]	Case Report	To describe the use of guided endodontics in calcified maxillary lateral incisor using small diameter bur.	Calcified maxillary left lateral incisor	SN	Not mentioned	Not mentioned	This method demonstrated an ultraconservative, highly reliable, and successful treatment without the excessive removal of enamel and dentin.
Lara-Mendes et al. (2018) [57]	Case Report	To describe an endodontic treatment technique performed through a new minimally invasive approach that leads to no tooth damage at the incisal edge and uses CBCT imaging and 3D guides.	Calcified maxillary left central incisor	SN	1 year	**Clinical:** Asymptomatic patient. **Radiographic:** Small alteration in the periodontal ligament space, which may be a sign of the presence of scar tissue.	The guided endodontic therapy optimized the treatment, having provided a conservative access with no tooth damage at the incisal edge in a safe and predictable way despite the presence of a severely calcified root canal.
Loureiro et al. (2021) [58]	Case Report	To discuss the impact of new technologies on the treatment of a complex case of a maxillary central incisor with PCO, lateral perforation and apical periodontitis treated using guided endodontics.	Calcified maxillary left central incisor	SN	6 months	The periapical radiograph and CBCT scan showed that the guided endodontics approach was radiographically successful.	CBCT contributed effectively for the diagnosis of a perforation that occurred during failed attempts to access the root canal. Guided endodontics, a quick, safe and comfortable intervention, is also very affordable because it does not require any surgical procedures.
Krastl et al. (2016) [59]	Case Report	To present a new treatment approach for teeth with PCO which require root canal treatment.	Calcified maxillary right central incisor	SN	15 months	**Clinical:** Asymptomatic with no pain on percussion **Radiographic:** No apical pathology	The presented guided endodontic approach seems to be a safe, clinically feasible method to locate root canals and prevent root perforation in teeth with PCO.
Hedge et al. (2019) [60]	Case Report	To report the management of PCO of maxillary central incisor using guided endodontic therapy.	Calcified maxillary right central incisor	SN	Not mentioned	Not mentioned	Guided endodontic therapy was found to be a time-saving method of the management of PCO in anterior teeth.
van der Meer et al. (2016) [61]	Case Report	To describe the application of 3D digital mapping technology for predictable navigation of obliterated canal systems during root canal treatment to avoid iatrogenic damage of the root.	Anterior teeth/ Not mentioned	SN	Not mentioned	Not mentioned	The method of digital designing and rapid prototyping of endodontic guides allows for reliable and predictable location of root canals of teeth with calcifically metamorphosed root canal systems.
Torres et al. (2019) [62]	Case Report	To describe a minimally invasive method to create a 3D-printed guide to gain access to obliterated root canals on the basis of CBCT data.	Calcified maxillary left lateral incisor	SN	6 months	**Radiographic:** Apical area completely healed.	Microguided Endodontics concept may be a valuable tool for the negotiation of PCO, reducing chair time and risk of iatrogenic damage to the root.
Torres et al. (2021) [63]	Case Report	To present a novel guided endodontics technique using a sleeveless 3D printed guide.	Calcified first maxillary right premolar	SN	1 year	**Radiographic:** Apical area completely healed.	This technique seems to be a promising alternative in comparison to the conventional guided endodontic guide-design for the negotiation of PCO in cases where vertical space is limited.
Buchgreitz et al. (2019) [64]	Case Report	To show a modification of guided endodontics with the purpose of reducing the need for interocclusal space using an intracoronal guide technique whereby it can be applied more often in the posterior region.	Calcified first maxillary right molar	SN	2 years	**Clinical:** Any objective symptoms of apical inflammation. **Radiographic:** Sustained lamina dura indicative of healing.	The demand for more interocclusal space was solved by transforming the virtual drill path into a composite-based intracoronal guide.
Lara-Mendes et al. (2018) [65]	Case report (2 treated teeth)	To describe a guided endodontic technique that facilitates access to root canals of molars presenting with pulp calcifications.	Calcified maxillary left second and third molars	SN	3 months	**Radiographic:** Regression of the periradicular lesions.	The guided endodontic technique in maxillary molars was shown to be a fast, safe, and predictable therapy and can be regarded as an excellent option for the location of calcified root canals, avoiding failures in complex cases.
					1 year	**Clinical:** Reduction of pain symptoms and a negative response to the percussion tests. **Radiographic:** Reduction of the periradicular lesions.	
Connert et al. (2018) [66]	Case report (2 treated teeth)	To present a novel miniaturized and minimally invasive treatment approach for root canal localization in mandibular incisors with PCO and apical periodontitis.	Calcified mandibular central incisors	SN	Not mentioned	Not mentioned	The ‘Microguided Endodontics’ technique is a safe and minimally invasive method for root canal location and prevention of technical failures in anterior teeth with PCO.
Krug et al. (2020) [45]	Case report (6 treated teeth)	To report the outcome of guided endodontic treatment of a case of dentin dysplasia with PCO and apical periodontitis based on the use of a 3D-printed template designed by merging CBCT and surface scan data.	Maxillary right lateral incisor, maxillary right second premolar, maxillary left first molar, mandibular left incisors and mandibular right first molar with dentin dysplasia	SN	1 year	**Clinical:** Asymptomatic teeth, with mobility improvement. **Radiographic:** Signs of apical lesion size reduction in teeth 36, 32 and 12. Complete healing of apical periodontitis obtained in teeth 15, 26, 31 and 46.	In patients with dentin dysplasia, conventional endodontic therapy is challenging. Guided endodontic treatment considerably facilitates the root canal treatment of teeth affected by dentin dysplasia.
Maia et al. (2019) [30]	Case report (3 treated teeth)	To describe the endodontic treatment of 1 molar and 2 premolars with extremely calcified canals using the guided endodontics technique.	Calcified maxillary first left molar; calcified maxillary second left upper premolar; calcified maxillary second right premolar	SN	15 days	**Clinical:** All the patients asymptomatic.	Although extremely detailed planning is required, the execution of the technique is relatively fast and safe, substantially reducing the occurrence of iatrogenic failures and increasing the success rates of endodontic treatment.
					6 months	**Radiographic:** **Case 1 and 3**–Periapical tissue mineralization. **Case 2**–Absence of periapical thickening.	
					1 year	**Radiographic:** ** Case 1, 2 and 3**–Complete healing	
Bordone et al. (2020) [67]	Case series (4 clinical cases)	To demonstrate the benefits and limits of static guided endodontics.	Calcified mandibular right canine; calcified maxillary right central incisor; calcified mandibular left canine; maxillary right canine	SN	Not mentioned	Not mentioned	SN assists endodontists in the management of complex cases by enabling centered drilling of the canal with minimum risk of deviating from the virtually planned path. The novel choice of a small-diameter drill (0.75 mm) helps maximize the preservation of the dental tissues.
Tavares et al. (2018) [7]	Case Series (2 clinical cases)	To describe 2 cases of guided endodontics programmed with conventional palatal access in anterior teeth and to discuss the applicability of this approach in cases of PCO with apical periodontitis and acute symptoms.	Calcified maxillary right central incisor	SN	**Case report 1:** 15 days **Case report 2:** 30 days.	**Clinical:** Asymptomatic teeth.	The method demonstrated high reliability and permitted proper root canal disinfection expeditiously, without the unnecessary removal of enamel and dentin in the incisal surface.
Ali et al. (2019) [14]	Case series (2 clinical cases)	To describe the conservative endodontic treatment of a type II DI (dens Invaginatus) with guided endodontic approach and 3D printed surgical stents.	**Case report 1:** maxillary incisors with type II DI **Case report 2:** maxillary right incisors with type II DI	SN	Not mentioned	Not mentioned	This technique provides a precise and minimally invasive approach in the conservative management of DI, without compromising the vitality of main pulpal tissue.
Llaquet et al. (2020) [36]	Case series (7 clinical cases)	To describe the endodontic management of seven teeth with PCO by means of digitally guided endodontics using both a virtually designed 3D guide and a customized 1 mm-diameter cylindrical bur.	Calcified maxillary left central incisor (3 case reports); calcified maxillary right canine (1 case report); calcified maxillary right central incisor (3 case reports)	SN	1 year	**Clinical:** No patients reported discomfort upon percussion or palpation at least one year after the treatment, when periapical healing was evident.	This treatment approach was demonstrated to be safe and fast and can be considered as a predictable technique for the location of calcified canals, thus minimizing complications.
Tavares et al. (2020) [33]	Case series (3 clinical cases)	To report a case series and describe the use of guided endodontics in complex symptomatic cases of mandibular and maxillary molars; presenting calcification of all three root canals.	Calcified mandibular second right molar; calcified mandibular right first molar; calcified maxillary right first molar	SN	**Case report 1: ** 15 days; 12 months **Case report 3:** 1 week; 12 months	**Clinical:** asymptomatic teeth.	The use of guided endodontics in cases of calcification in molars was demonstrated to be a viable and reliable alternative treatment.
Shaban et al. (2021) [68]	Case series (2 clinical cases)	To describe the role of guided endodontics with ultrasonic tips in management of calcified canals.	Calcified maxillary left central incisor; calcified maxillary right first molar	SN	**Case report 1 and 2:** 2 weeks	**Clinical:** Asymptomatic, negative on palpation and percussion.	Endodontics guides with ultrasonic tips are reliable in management of root canal calcifications. Three-dimensional imaging using CBCT and CAD/CAM provides accurate 3D guides.
					**Case report 1 and 2:** 1 year	**Radiographic:** Normal periapical radiographic appearance.	

3D: 3 dimensional; CBCT: cone-beam computed tomography; DI: dens invaginatus; PCO: pulp canal obliteration; SN: computer-aided static navigation.

## Data Availability

Not applicable.

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
