# Peer review of "Guided Endodontics: Static vs. Dynamic Computer-Aided Techniques—A Literature Review"

_jpm, 2022, doi:10.3390/jpm12091516_

Round 1
Reviewer 1 Report
I express a favorable opinion for the publication of this article. Readers will be able to find valid references for studying complex cases in endodontics using suggestions from selected articles.
Author Response
Reviewer #1
- I express a favorable opinion for the publication of this article. Readers will be able to find valid references for studying complex cases in endodontics using suggestions from selected articles.
Author’s response: We thank the reviewer #1 for the valuable comments and feedback.
Revised text: Not applicable.
Reviewer 2 Report
Dear Editor,
The paper presents a literature review which compared static and dynamic computer-aided techniques. The authors have put a great effort into drafting this article and discussed topic is interesting.
As a reviewer, I have some concerns:
1. Various cavity accesses appear in the table and article text. Please describe briefly and concisely the possible access cavity designs.
2. Organize table Table 1. Included original research studies. I suggest that the first should be studies evaluated SN, next DN, finally SN vs DN.
3. Please provide in the results section how many of selected research concern only SN, only DN, and SN vs DN?
Author Response
Reviewer #2
- Dear Editor,
The paper presents a literature review which compared static and dynamic computer-aided techniques. The authors have put a great effort into drafting this article and discussed topic is interesting.
Author’s response: We want to thank reviewer #1 for the opportunity to revise the manuscript and carefully proceed with the required modifications.
Revised text: Not applicable.
- As a reviewer, I have some concerns: Various cavity accesses appear in the table and article text. Please describe briefly and concisely the possible access cavity designs.
Author’s response: Following the reviewer #1 valuable suggestion, the access cavity designs topic has been addressed in theIntroduction section.
Revised text: “It is well-known that access cavities should be kept as conservative as possible, following the trend of minimally invasive dentistry. The achieved tooth preservation may be an efficient way to decrease the occurrence of post-treatment tooth fractures (16). Access cavity preparations involving both mesial and distal marginal ridges can reduce cuspal stiffness by up to 63% [16]. Although nomenclature in science is crucial to convey ideas and concepts unambiguously, the standardization of access cavity terminology is still an issue. Generally, whether streaming from social media or purely scientific lines, approaches such as “ninja” and truss cavities that preserve the dentinal bridge by producing two or more occlusal access multi-rooted teeth have flourished. The most widely accepted concepts for access cavities are the traditional, the conservative with parallel or divergent walls, the ultra-conservative, the caries-driven, the restorative-driven and the truss cavity (16).” - Introduction section, page 2, lines 60 to 71.
- Organize table Table 1. Included original research studies. I suggest that the first should be studies evaluated SN, next DN, finally SN vs DN.
Author’s response: Once again we thank the reviewer #1 for the remark. Table 1 has been altered following the reviewer’s suggestion.
Revised text: Results section, page 5 to 11, Table 1.
- Please provide in the results section how many of selected research concern only SN, only DN, and SN vs DN?
Author’s response: We thank the reviewer #1 for the feedback which objective was to enhance the quality of the manuscript. The requested information was added.
Revised text: “Fourteen (14) original research studies focused on computer-aided static technique (refs) and 9 on the dynamic navigation method (refs). Only one study comparatively evaluated both guided endodontics’ techniques (refs). Moreover, 24 case reports and 6 case series assessed static technique’s clinical and/or radiographic outcomes.” - Results section, page 3, lines 134 to 137.